# Constraining Young Hot Jupiter Occurrence Rate in Stellar Associations Using 2-min Cadence TESS Data

**Yuanqing Fang** [1,2], **Bo Ma** [1,2,*], **Chen Chen** [2,3] **and Yongxin Wen** [1,2]

1    Department of Astronomy, School of Physics and Astronomy, Sun Yat-sen University, Zhuhai 519082, China
2    CSST Science Center in the Guangdong-Hongkong-Macau Great Bay Area, Sun Yat-sen University, Zhuhai 519082, China; omvmjs@gmail.com
3    School of Statistics and Data Science, Zhuhai College of Science and Technology, Zhuhai 519082, China
*    Correspondence: mabo8@mail.sysu.edu.cn

**Abstract:** The characterization of young planet distribution is essential for our understanding of the early evolution of exoplanets. Here we conduct a systematic search for young planets from young open clusters and associations using the 2-min cadence TESS survey data. We obtain TESS light curves for a total of 1075 young stars, which are selected with the aid of Gaia data. There are a total of 16 possible transiting signals. After a thorough vetting process, some have been confirmed as planets, and others are likely caused by eclipsing binaries. The final sample contains six confirmed planets, of which one is a hot Jupiter. After accounting for survey completeness using a Monte Carlo simulation, we can put a 95% confidence level upper limit on the hot Jupiter (P < 10 days, Rp = 0.7–2.9 $R_{Jup}$) occurrence rate orbiting stars in young associations at <5.1% and a 68% confidence level upper limit at <2.5%. We estimate that a sample size of ∼5000 dwarf stars with 2-min cadence data will be needed to reach a 0.5% upper limit on the hot Jupiter occurrence rate, which is the typical hot Jupiter occurrence rate around main sequence stars. Thus, future studies with larger sample sizes are required to put more constraints on planet formation and evolution theories.

**Keywords:** exoplanet; young star; transiting planet

---





## 1. Introduction

Though lots of exoplanets have been discovered, the sample of young planets (age ≤ 1 Gyr) is still relatively small. The demographics of planets around young (<1 Gyr) stars is a crucial link toward a more direct comparison with planet formation and evolution theories [1,2]. In their early evolution stage, migration, photoevaporation, mass loss, and dynamical interaction with other planets can occur and sculpt the demographics of the planet population. Thus, detecting young planets can provide valuable information for our understanding of planet formation and evolution, since the first few hundred million years of planet evolution define the planetary systems we observe today.

However, it is usually very difficult to detect planets around young and active stars using radial velocity technique without special treatment of their activity signal [3–7]. Thus, lots of efforts have been put into searching for transiting exoplanets around young stars [8–15]. Some recent discoveries are listed below. Mann et al. [8] started a K2 campaign and confirmed a close-in super-Neptune planet transiting a pre-main sequence (11-Myr-old) star in the Upper Scorpius OB association, which suggests close-in planets can either form in situ or have finished migration within ∼10 -Myr. Mann et al. [10] identified seven planet candidates in the Praesepe Cluster and saw increasing evidence that some planets continue to lose atmosphere past 800 Myr. Mann et al. [13] validated TOI 1227 b, a $0.85 \pm 0.05$ $R_J$ planet transiting a very low-mass star ($0.170 \pm 0.015$ $M_\odot$) every 27.4 days, and suggested that TOI 1227 b is still contracting. Sun et al. [15] searched for planet candidates in young stellar associations and open clusters based on the TESS Object of Interest Catalog. They found one confirmed planet, one promising candidate, one brown dwarf, three unverified

planet candidates in open clusters, and ten planet candidates in young stellar associations. Most of the previous studies focus on targets in young cluster or association. However, Zhou et al. [14] and Desidera et al. [16] have also found young planetary systems around field stars using the TESS survey data. The young stellar ages are inferred using several stellar activity indicators. Dai et al. [17] detected six transiting planets with radii between 2-5$R_\oplus$ in a young planetary system (700 Myr) around TOI-1136.

Hot Jupiters are gas giants with orbital periods shorter than 10 days and masses greater than or equal to 0.25 Jupiter masses [18]. Johnson et al. [19] found only 1 percent of sun-like stars host one hot Jupiter, and the occurrence rate falls off around the M dwarfs. In the Kepler survey, the occurrence rate of hot Jupiters (P < 10 days, Rp = 0.7–2.9 $R_{Jup}$) is $0.4 \pm 0.1\%$ for GK Dwarfs with Kp < 15 mag [20]. Deleuil et al. [21] obtained a value of $0.98 \pm 0.26\%$ using CoRoT survey results. It is still not clear yet whether this occurrence rate will evolve from young stellar systems to old mature stellar systems, which is critical for our understanding of the formation and evolution of hot Jupiters [15,18,22] and the brown dwarf desert [23–25].

There are several competing hot Jupiter formation scenarios, including in situ formation, gas disk migration, and high-eccentricity tidal migration (please see [18] and references therein). Measuring the change in hot Jupiter occurrence rate with host star age may help distinguish among the origin scenarios. For example, if high-eccentricity tidal migration is at work, the occurrence rate of hot Jupiters will tend to be lower in young clusters since the time scale for this process requires hundreds of Myr up to 1 Gyr for the completion [26]. Several studies have been carried out to measure the occurrence rate of young, hot Jupiters using the transit method. Early transit surveys targeting open clusters typically yield a high Jupiter occurrence rate with very large uncertainties due to their less than ideal observing cadence and photometry precision, such as <24% and <25% [27,28]. By analyzing the 30-min cadence TESS full frame image, Nardiello et al. [29] obtained a rate of $0.19 \pm 0.07\%$ for targets in the open clusters of the southern ecliptic hemisphere, which is smaller than what was estimated for field stars [30]. However, the lack of a completeness study and the false positive rate estimate make this result still provisional [29].

Here we propose to use the 2-min cadence TESS survey data, with its relatively long time baseline and high photometry precision, to search for short-period young exoplanets and constrain the occurrence rate of young hot Jupiters in young stellar associations. This paper is organized as follows. In Section 2, we will present the data used in this study. Section 3 will present our search method and results. We will put an upper limit on the hot Jupiter occurrence rate in Section 4 and give a summary in Section 5.

## 2. Data and Observation

The Transiting Exoplanet Survey Satellite (TESS) is an all-sky survey mission targeting planets around bright stars [31]. It is equipped with 4 cameras and observes a small fraction of the sky during each sector. Each sector lasts about an average of 27 days and covers 24° × 96° sky area. The camera has a pixel scale of about 21 arcsec. It began the all-sky survey in 2018 July and the survey is still ongoing.

Since it is conducting an all-sky survey, TESS also sees a large number of known young open clusters and associations. Thus we can obtain high precision photometry data for a large number of stars in these clusters/associations and use these data to search for young transiting planet signals. We select clusters/associations with ages spreading from 10 to about 1 Gyr. The young clusters or associations we searched are listed in Table 1. Their ages and distances are obtained from Banyan-Σ in Gagné et al. [32], Gaia Collaboration et al. [33] (NGC 2451A, $\alpha$ Persei, Blanco 1), Gagné et al. [34] (Volans-Carina) and Hawkins et al. [35] (Pisces-Eridanus).

We then proceed to obtain the member lists of these young associations from literature [32,34,36–40] and cross-match them with the TESS database. The majority of these member lists are compiled by constructing a model using kinematic information and astrometry data. The stellar masses, radii, and contamination ratios are adopted from

the Tess Input Catalog [41,42]. We only select targets with the 2-min cadence TESS data. Similar to the study of Nardiello et al. [29], we have removed targets that have $R_\star > 2\,R_\odot$ to concentrate on solar-type stars. In the end, we find a total of 1075 targets from young clusters/associations with TESS observation. We would expect a small percentage (on the order of 10%) of incorrect members from these lists, which will result in a 10% fractional error on our final occurrence rate estimation. We present the color-magnitude diagram of all these stars in Figure 1 based on Gaia DR3 data, where the MESA Isochrones and Stellar Tracks (MIST) isochrones are overplotted for reference [43,44]. When making the color-magnitude diagram, we adopt the correction to photometry of individual targets from interstellar extinction using the Gaia DR3 catalog values of $A_G$ and $E(BP - RP)$ [45]:

$$M_G = G + 5\log\pi + 5 - A_G \qquad (G_{\rm BP} - G_{\rm RP})_0 = G_{\rm BP} - G_{\rm RP} - E(G_{\rm BP} - G_{\rm RP}), \qquad (1)$$

where $G$ is the apparent magnitude and $A_G$ is the interstellar extinction, $\pi$ the parallax, $E(G_{\rm BP} - G_{\rm RP})$ the color excess. We have used the MIST web interface to interpolate isochrones at 10 and 100 million years, assuming solar metallicity. The number of available targets with TESS 2-min light curve data in each cluster/association is also summarized in Table 1. Double or multiple stars are treated as one stellar system here.

**Table 1.** Number of targets analyzed in this work. We show the number of targets analyzed using TESS 2-min cadence data from young clusters and associations in this work. We list the young clusters and associations by the order of their ages, which are adopted from literature works shown in the reference column.

| Cluster/Association | Distance (pc) | Age (Myr) | Targets Number | Reference |
|---|---|---|---|---|
| Upper CrA | $147 \pm 7$ | $\sim 10$ | 2 | [32] |
| Upper Scorpius | $130 \pm 20$ | $10 \pm 3$ | 4 | [46] |
| $\eta$ Chamaeleontis | $95 \pm 1$ | $11 \pm 3$ | 9 | [47] |
| Lower Centaurus Crux | $100 \pm 10$ | $15 \pm 3$ | 83 | [46] |
| Upper Centaurus Lupus | $130 \pm 20$ | $16 \pm 2$ | 70 | [46] |
| 32 Orionis | $96 \pm 2$ | $22^{+4}_{-3}$ | 6 | [47] |
| $\beta$ Pictoris | $30^{+20}_{-10}$ | $24 \pm 3$ | 39 | [47] |
| Octans | $130^{+30}_{-20}$ | $35 \pm 5$ | 7 | [48] |
| Columba | $50 \pm 20$ | $42^{+6}_{-4}$ | 19 | [47] |
| Carina | $60 \pm 20$ | $45^{+11}_{-7}$ | 5 | [47] |
| Tucana-Horologium | $46^{+8}_{-6}$ | $45 \pm 4$ | 55 | [47] |
| IC 2602 | $146 \pm 5$ | $46^{+6}_{-5}$ | 38 | [49] |
| IC 2391 | $149 \pm 6$ | $50 \pm 5$ | 14 | [50] |
| NGC 2451A | $\sim 193$ | $\sim 60$ | 25 | [51] |
| Platais 8 | $130 \pm 10$ | $\sim 60$ | 5 | [52] |
| $\alpha$ Persei | $\sim 176$ | $60 \pm 7$ | 89 | [53] |
| Volans-Carina | $75$–$100$ | $89^{+5}_{-7}$ | 10 | [34] |
| Blanco 1 | $236.4$ | $\sim 100$ | 156 | [51] |
| Pleiades | $134 \pm 9$ | $112 \pm 5$ | 113 | [54] |
| Pisces-Eridanus | $80$–$226$ | $120$ | 146 | [35] |
| AB Doradus | $30^{+20}_{-10}$ | $149^{+51}_{-19}$ | 48 | [47] |
| Carina-Near | $30 \pm 20$ | $\sim 200$ | 80 | [55] |
| UMa | $25.4^{+0.8}_{-0.7}$ | $414 \pm 23$ | 61 | [56] |
| XFOR | $100 \pm 6$ | $\sim 500$ | 6 | [57] |
| Coma Ber | $85^{+4}_{-5}$ | $562^{+98}_{-84}$ | 85 | [53] |

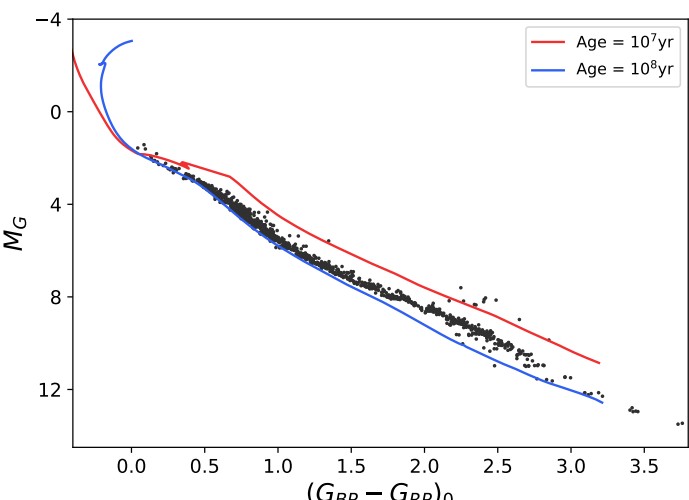

**Figure 1.** Gaia Hertzsprung-Russell diagram for all our survey targets using Gaia DR3 data. $M_{\mathrm{G}}$ denotes the absolute Gaia broadband magnitude after extinction correction and $(G_{\mathrm{BP}} - G_{\mathrm{RP}})_0$ the Gaia color after color excess correction. For reference, the interpolated isochrones from MIST website are also overplotted.

In order to obtain a high signal-to-noise ratio (SNR) transit detection and better understand the architecture of each system, we make use of all available 2-min cadence TESS light curve processed by the Science Processing Operations Center (SPOC, Jenkins et al. [58]) for our stellar sample. We retrieve the light curves of each target from Mikulski Archive for Space Telescopes (MAST2) via astroquery [59]. Most of the data are collected between 2018 and 2022 by TESS, which are mainly in sector 1–45.

## 3. Methods and Results

### 3.1. Candidate Search and Vetting

Based on Python, Lightkurve package provides tools to analyze time series data on the brightness of planets, stars, and galaxies, mainly for Kepler and TESS data [60]. We first use Lightkurve to flatten and reject outliers from the light curves. Then we use Wōtan, a Python package developed to detrend light curves [61], to remove the stellar variability typically seen in light curves of young stars. We decided to use Tukey's biweight filter and a running window of three times the duration of a central transit on a circular orbit at any given trial orbital period to detrend the light curve from long-term stellar variability and from any systematic trends, which has been shown to yield the highest recovery rates of injected transits from simulated data [61].

The resulting light curves are suitable for the purpose of searching transiting signals. We mainly use the Box-fitting Least Squares (BLS, Kovács et al. [62]) and Transit-Fitting Least Squares (TLS, Hippke and Heller [63]) periodogram tool in Lightkurve to search for periodic transiting or eclipsing signals from the light curves. We require (i) at least two transits, (ii) an SNR > 7.1, and (iii) a signal detection efficiency [62] SDE_TLS > 9 for a transiting event to be identified as a possible transiting signal.

All possible transiting signals are then examined after we phase-fold the light curves, where the following vetting tests are taken into account: checking for secondary eclipses, comparing the depths of odd/even transits, and checking for contamination from nearby stars. We will force the TLS package to fit the light curve data at a window near the expected secondary eclipse (phase between 0.85 to 1.15 to allow for mild eccentric orbit) with a fixed period of the candidate. We require the fitted secondary eclipse depth to be smaller than 10% of the primary depth to pass the secondary eclipse test [64].

For the purpose of checking odd and even transit depths, we utilized the TLS package. This package provides outputs for both odd and even depths. We require the median odd-even transit depth difference to be smaller than 3 times the fit uncertainties of odd and even transit depth to pass the test:

$$|\delta_{even} - \delta_{odd}| < 3 \times (\sigma^2_{\delta_{even}} + \sigma^2_{\delta_{odd}})^{1/2}, \tag{2}$$

where $\delta$ is the transit depth and $\sigma$ is the fitting uncertainty.

To check for possible contamination, we use the in/out-of-transit centroid difference test [65] to check if the transit events are associated with the target or with a nearby background star. The centroid positions are taken from the SPOC data products. To conduct the test, the 3-h lightcurve data immediately before and after the transit is selected to serve as the out-of-transit data. A smooth linear trend is fitted using the out-of-transit data and removed from all centroid data [66]. A Student *t*-test is used to test whether the means of two distributions are consistent. We chose a probability *p* value of 5% for the *t*-test, and the centroid positions inside and outside transits are not considered to come from the same star. The flow chart of the searching process is shown in Figure 2.

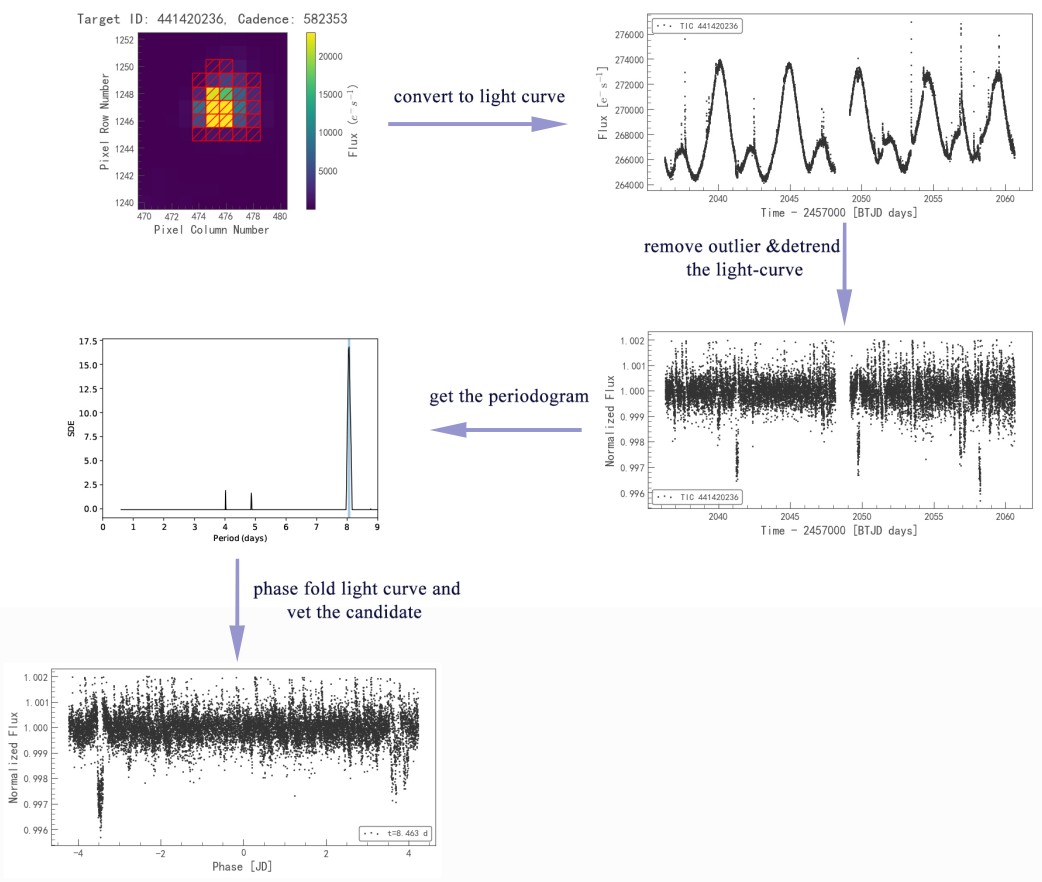

**Figure 2.** Flow chart of the young planet search process. We use TIC 441420236 as an example here.

We have identified a total of 16 possible transit signals in the whole sample using our pipeline, which are summarized in Table 2. Among them, six candidates have been previously identified as transiting planets and six as eclipsing binaries, which are presented in Sections 3.2 and 3.4 (please also see Table 3). There are another four possible transit signals that have not been previously studied and that warrant further analysis in Section 3.3. The mass and radius of possible planetary host stars are summarized in Table 4.

**Table 2.** The transit/eclipsing signals detected from the sample.

| Source | Association/Cluster | Description |
|---|---|---|
| AU Mic | BPMG | Confirmed planet [1] |
| DS Tuc A | THA | Confirmed planet [2] |
| TOI-837 | IC2602 | Confirmed planet [3] |
| HD 63433 | UMa | Confirmed planet [4] |
| Gaia DR2 5536809162106730112 | NGC2451 | likely EB |
| HD 20701 | Alpha Per | background variable star |
| HD 224112 | Blanco 1 | background EB |
| 2MASS J00024841-2953539 | Blanco 1 | eclipsing binary [5] |
| V* V1283 Tau | Pleiades | eclipsing binary [6] |
| RS Cha | ETAC | eclipsing binary [7,8] |
| CD-46 9495 | UCL | eclipsing binary [9] |
| GG Lup | UCL | eclipsing binary [10] |
| BS Ind | THA | eclipsing binary [11] |

Reference: [1] Martioli et al. [67], [2] Newton et al. [68], [3] Bouma et al. [2], [4] Mann et al. [12], [5] Smith et al. [69], [6] Barros et al. [70], [7] Cousins [71], [8] Steindl et al. [72], [9] Kiraga [73], [10] Clausen et al. [74], [11] Szczygieł et al. [75].

**Table 3.** Properties of confirmed exoplanets.

| Planet | Period (Day) | Mass ($M_{\mathrm{Jup}}$) | Radius ($R_{\mathrm{Jup}}$) | Reference |
|---|---|---|---|---|
| AU Mic b | 8.463 | 0.054 | 0.371 | [67] |
| AU Mic c | 18.859 | 0.046 | 0.295 | [67] |
| DS Tuc Ab | 8.138 | <0.045 | 0.508 | [68,76] |
| TOI-837 b | 8.325 | <1.2 | 0.77 | [2] |
| HD 63433 b | 7.11 | - | 0.19 | [12] |
| HD 63433 c | 20.55 | - | 0.24 | [12] |

**Table 4.** Properties of possible planet host stars.

| Star | Mass ($M_{\odot}$) | Radius ($R_{\odot}$) | Reference |
|---|---|---|---|
| AU Mic | 0.5 | 0.75 | [77] |
| DS Tuc A | 0.959 | 0.872 | [78] |
| TOI-837 | 1.118 | 1.022 | [2] |
| HD 63433 | 0.99 | 0.912 | [12] |
| Gaia DR2 5536809162106730112 | - | 0.74 | [79] |
| HD 20701 | 1.97 | 1.85 | [80] |
| V* V1283 Tau | - | 0.71 | [79] |

### 3.2. Confirmed Planetary System

During our search, we have independently detected four previously known exoplanetary systems, which include AU Mic, DS Tuc A, TOI-837, and HD 63433.

**AU Mic** As a member of $\beta$ Pictoris, the star has a mass of $\sim$0.5 $M_{\odot}$ and radius of $\sim$0.75 $R_{\odot}$. It was observed by TESS during Sector 1 and 27. The raw light curve shows a modulation with a semi-amplitude of $\sim$2% and a period of $\sim$4.9 days. The detrended and phase folded light curves are shown in Figure 3, where we have identified the transit signals of AU Mic b and AU Mic c. Plavchan et al. [77] first reported the detection of planet AU Mic b, which has a period of $\sim$8.5 days. After analyzing more TESS observation data, Martioli et al. [67] then reported the detection of another Neptune-sized planet in this system, AU Mic c, with a period of $\sim$19 days. From the TESS light curve, the star is seen in a relatively active phase during the TESS observation, where outbursts from the host stars are clearly seen.

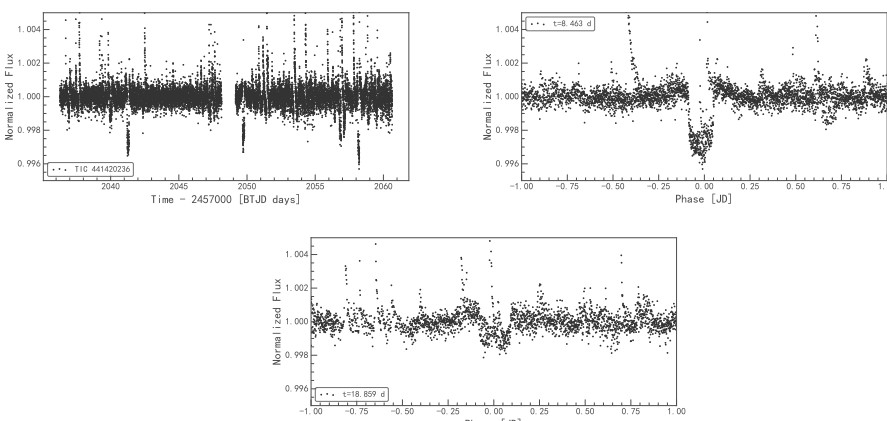

**Figure 3.** TESS light curve of AU Mic (Sector 27). (**Top left**): the detrended light curve. (**Top right** and **Bottom**): the phase-folded light curve to the period of 8.463 d and 18.859 d, respectively.

**DS Tuc A** The star is a member of the Tucana-Horologium young moving group, which has a mass of $\sim$1 $M_\odot$ and radius of $\sim$0.9 $R_\odot$. It was observed by TESS in Sectors 1, 27, and 28. The raw light curve shows a modulation with a semi-amplitude of $\sim$3% and period of $\sim$2.9 day. The detrended and phase-folded light curves are shown in Figure 4. Newton et al. [68] reported the detection of DS TUC Ab in this system, which is a planet with size larger than Neptune but smaller than Saturn and a period of 8.14 day. From the TESS light curve, the star is in a relatively active phase during the TESS observation, which makes it relatively harder to search for smaller planets.

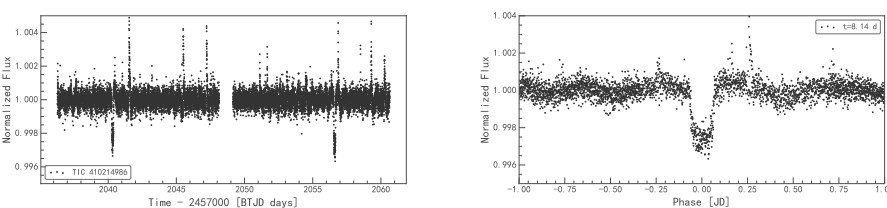

**Figure 4.** TESS light curve of DS Tuc A (Sector 27). (**Left**): the detrended light curve. (**Right**): the phase-folded light curve to the period of 8.14 days.

**TOI-837** TOI 837 is a member of open cluster IC 2602, the Southern Pleiades. It has a mass of $\sim$1.1 $M_\odot$ and radius of $\sim$1 $R_\odot$. It was observed by TESS in Sector 10 and 11. The raw light curve shows a modulation with an semi-amplitude of $\sim$1% and period of $\sim$3 day. The detrended and phase folded light curves are shown in Figure 5. From the TESS ligexamining thht curve, the star is in a relatively quiet phase during the TESS observation, which makes it relatively easier to search for smaller planets. Bouma et al. [2] detected one planet slightly smaller than Jupiter in this system, with a period of 8.325 days.

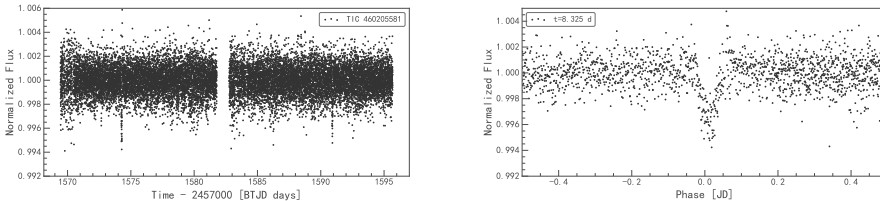

**Figure 5.** TESS light curve of TOI-837 (Sector 10). (**Left**): the detrended light curve. (**Right**): the phase-folded light curve to the period of 8.325 days.

**HD 63433** This star is a member of the UMa moving cluster. It has a mass of $\sim$1 $M_\odot$ and a radius of $\sim$0.9 $R_\odot$. It was observed by TESS during Sector 20. The raw light curve shows a

modulation with a semi-amplitude of ∼1% and a period of ∼6.5 day. The detrended and phase folded light curves are shown in Figure 6. Mann et al. [12] reported the detection of two planets in this system, with periods of 7.11 and 20.55 days respectively. By examining the TESS light curve, we see that the star is in a relatively quiet phase during the TESS observation, which makes it relatively easier to detect smaller planets.

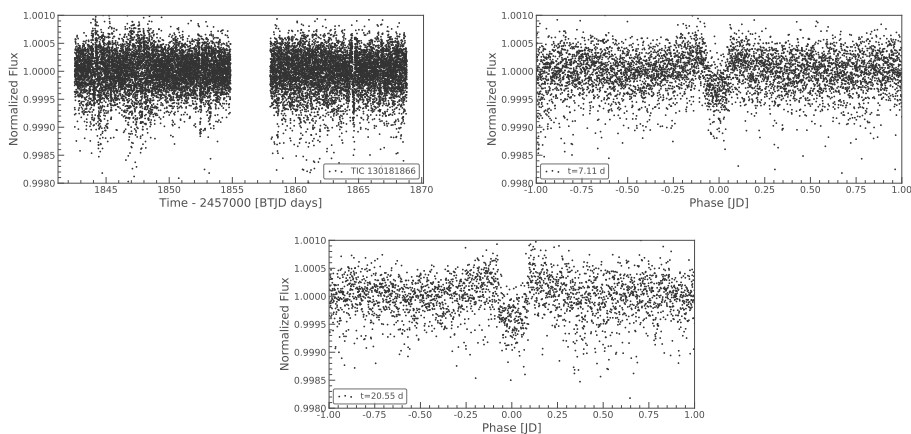

**Figure 6.** TESS light curve of HD 63433 (Sector 20). (**Top Left**): the detrended light curve. (**Top right** and **Bottom**): the phase-folded light curve to the period of 7.11 d and 20.55 d, respectively.

### 3.3. False Positive Transiting Signals

In this section, we will report the vetting results for four possible transiting signals from our pipeline.

**Gaia DR2 5536809162106730112** The target is a member of the open cluster NGC 2451, which has a radius of 0.74 $R_{\odot}$. It was observed by TESS in Sectors 34 and 35. The detrended and phase folded light curves are shown in Figure 7. The transit depth is about 8%, with a V-type shape. During the vetting process, we find that the odd and even numbers of transit depths differ by ∼100 ppm. No contamination from nearby source is detected, which suggests this system is likely a grazing binary with a period of 5.56 days, or has a 5.56 -day eclipsing binary in its background.

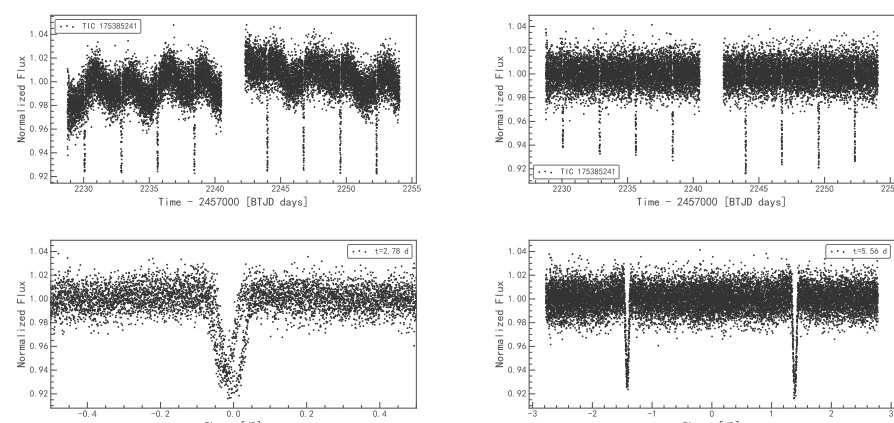

**Figure 7.** TESS light curve of Gaia DR2 5536809162106730112 (Sector 34). From top to bottom, light curve before detrending, after detrending, phase-folded to the period of 2.78 d, and to the period of 5.56 d.

**HD 20701** The target is a member of the open cluster Alpha Per, which has a mass of 2 $M_{\odot}$ and radius of 1.85 $R_{\odot}$. It was observed by TESS in its Sector 18. The detrended and phase folded light curves are shown in Figure 8. We find a 0.90 -day transit signal from this

system. We find the transit depth is changing when we change the aperture mask used. This suggests that the transit signal doesn't come from the target star, and may come from a variable star in the background.

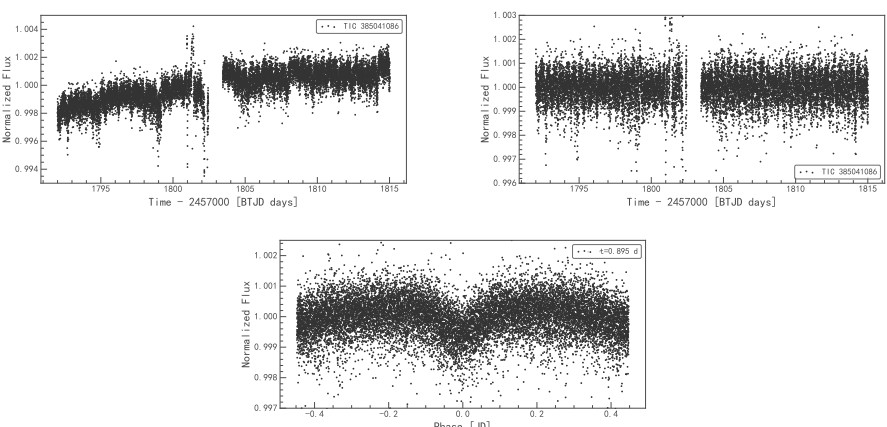

**Figure 8.** TESS light curve of HD 20701 (Sector 18). From top to bottom, we show light curves before detrending, after detrending, and phase-folded to the period of 0.895 d.

**HD 224112** This star is a member of the open cluster Blanco 1. It was observed by TESS in Sector 2 and 29. The detrended and phase-folded light curves are shown in Figure 9. We detect a V-shaped transit signal with a depth of 0.2% and a period of 2.45 days. After further analysis, we find a deeper transit signal with the same period from a nearby star, V* AL Scl, which is a known eclipsing binary [81]. We conclude that the transit signal does not come from HD 224112 but is caused by contamination from a nearby eclipsing binary.

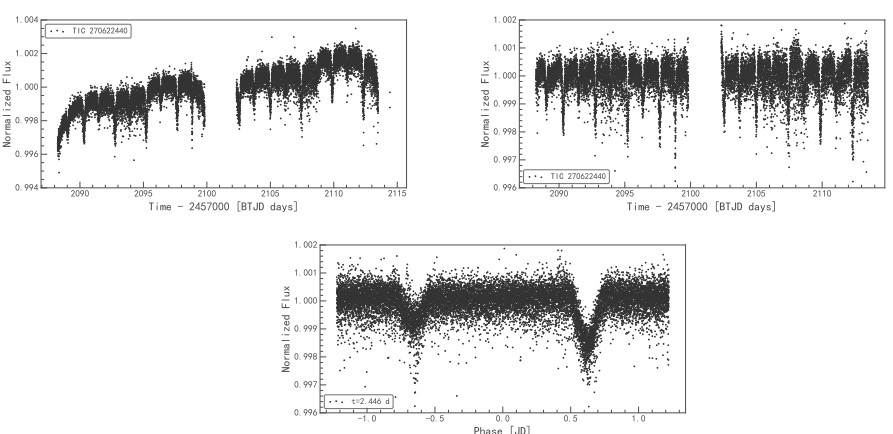

**Figure 9.** TESS light curve of HD 224112 (Sector 29). From top to bottom, we show light curves before detrending, after detrending, and phase-folded to the period of 2.446 d.

### 3.4. Previously Known EBs

Whenever we find a possible transiting or eclipsing signal in the searching process, we will check literature to see if this system has been previously studied. In total, we have found six previously known eclipsing binaries, which are summarized in Table 2 along with the discovering reference papers. These eclipsing binaries have a period in the range of 0.44 to 16.8 days, and an eclipsing depth between 1.5% and 42%. Once having radial velocity observation and mass measurements, these eclipsing binaries will become valuable targets for the purpose of calibrating the evolution of young stars.

## 4. Discussion

### 4.1. Search Completeness for Hot Jupiter

We have carried out an Monte Carlo study to determine our search completeness for hot Jupiters. Following Howard et al. [20], here we define hot Jupiters as planets with P < 10 days and Rp = 0.7–2.9 $R_{Jup}$.

We first measure the detection efficiency of our pipeline using a planet injection-recovery method. We inject planet signals into the SPOC raw light curves, then feed the simulated light curves to our detection pipeline. We uniformly divide the period-radius space into a 4 × 4 grid. In each period-radius bin, we randomly inject 5 periodic transit signals to each star in our sample without transit/eclipsing signals being detected. The planet radius $R^{inj}$ and orbital period $P^{inj}$ are drawn randomly from uniform distributions according to the parameters of the period-radius bin. Then we randomly generate the impact parameter $b$ from a uniform distribution between 0 and 1.0 and the mid-transit time $T_0$ from a uniform distribution between the light curve start time $t_0$ and $t_0 + P^{inj}$. The transit signal is calculated using the BAsic Transit Model calculation in Python (batman) code [82], assuming a fixed circular orbital eccentricity ($e = 0$), a quadratic limb-darkening model with fixed coefficients $(0.3, 0.3)$ for simplicity, and a contamination ratio value from the TESS Input Catalog.

We consider the injected planet as recovered if it passes our selection and vetting process as described in Section 3.1 and if the recovered period is within 10% of the injected period. The detection efficiency map is generated based on the fraction of recovered planets in each period-radius bin and is shown in Figure 10. The simulation shows we can reach a ~90% sensitivity when searching transiting hot Jupiters from our sample targets using the TESS 2-min cadence data.

We then correct for the geometric probability of transit to calculate the search completeness map. The geometric transit probability is defined as

$$p_{transit} = \frac{R_\star}{a}. \tag{3}$$

For each randomly injected planet, we compute the transit probability and multiply this factor intong hot Jup the average detectability map to account for the geometric effect. The final average completeness map is shown in Figure 10, with an average completeness of ~0.086.

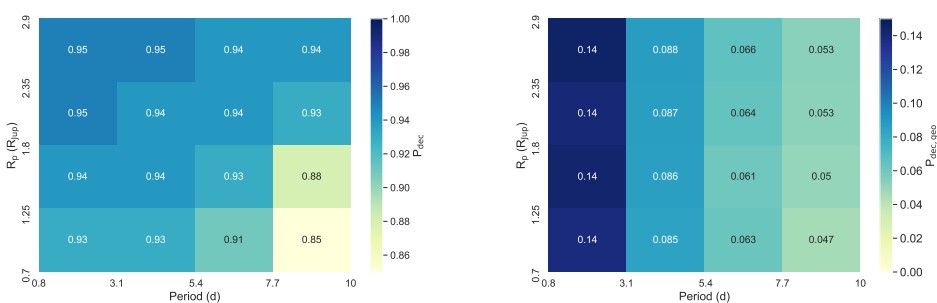

**Figure 10.** (**Left panel**): The average sensitivity of our detection pipeline as a function of orbital period and planet radius based on a injection-recovery simulation. (**Right panel**): The average search completeness map after accounting for the transit probability. Darker colors indicate higher numerical values.

### 4.2. Occurrence Rate of Hot Jupiter in Young Stellar Associations

One interesting question in exoplanet study is whether the occurrence rate of young, hot Jupiter will change with the age of the stellar system due to an evolutionary effect of the planets. We here try to constrain the occurrence rate of hot Jupiter in young stellar associations using results from this study. We have identified one possible transiting

hot Jupiter, TOI-837 b, out of a sample of 1075 young stars. For simplicity, we use the average survey completeness of 0.086 derived from the completeness map to constrain the occurrence rate. Based on the cumulative probability of the Poisson distribution, if there are 4.75 hot Jupiters expected to be detected from our survey, the probability of us detecting one hot Jupiter or fewer is 5% and the probability of us detecting more than one hot Jupiter is 95%. Therefore, we can conclude that the occurrence rate of hot Jupiters in young stellar associations is less than $4.75/1075/0.086 = 5.1\%$, with a *p*-value of 95%. Similarly, if there are 2.35 hot Jupiters expected to be detected from our survey, the probability of us detecting one hot Jupiter or fewer by us is 32% and the probability of detecting more than one hot Jupiter is 68%. Thus, the occurrence rate is less than $2.35/1075/0.086 = 2.5\%$, with a *p*-value of 68%.

Another factor that can impact our occurrence rate analysis is the inclusion of non-member stars/old stars in our sample. We adopt member lists from literature works, which often use dynamic models to assign certain stars to a particular star association. If we assume a conservative 10% chance that a member star does not belong to young associations [37], the upper limit of the occurrence rate would drop 10%, from 5.1% to 4.6%.

Our result agrees with the radial velocity survey of open cluster M 67, which gave a hot Jupiter occurrence rate of $5.7\%^{+5.5}_{-3.0}$ [83]. Another survey from Hartman et al. [27] gave a upper limit of <12% for 1.0 $R_{\rm Jup}$ planets with period <5 days. If young stars have a larger hot Jupiter occurrence rate than that of the main-sequence stars [19,20], then it means that a large fraction of hot Jupiters have been destroyed by their host stars during the early evolution stage, which is either driven by tidal force between star and planet or by dynamic interaction between multiple planets in the same system. Hamer and Schlaufman [84] use data from Gaia DR2 to show the population of hot Jupiter host stars is on average younger than the field population, which supports their claim that a lot of hot Jupiters are destroyed during the main sequence of their host star. This would lead to a lower occurrence rate for hot Jupiters around older stars. The most straightforward way to verify this scenario is to have a better measurement of the occurrence rate of young, hot Jupiters. Using the 30-min cadence TESS full frame image data, Nardiello et al. [29] obtained a rate of $0.19 \pm 0.07\%$ for targets in the open clusters. However, the lack of a completeness study and a false positive rate estimate make their result still preliminary. To reach an 0.5% upper limit on the hot Jupiter occurrence rate measurement using our method, a sample size of $\sim$5000 dwarf stars will be needed. With a planned second TESS Extended Mission spanning Years 5–7 on the horizon, our future work utilizing data from more stars can obtain a better constraint on the occurrence rate.

### 4.3. Distribution of Young Planets

Planets can evolve with time. Thus, it is possible to constrain the evolution theories of exoplanets by detecting young planets and comparing their distribution with that of Gyr-old mature planets [1,2]. We have shown all young planets from our sample in Figure 11, together with all known exoplanets from the NASA exoplanet archive database. We have limited the selection of known planet samples to planets with a fractional radii measurement error smaller than 30% to reduce the impact of outliers. First noticed by Bouma et al. [2], sub 100 Myr transiting planets do not overlap with the known populations of either hot Jupiters or sub-Neptune-sized planets in the period-radii diagram. Nardiello et al. [85] found a concentration of objects with $4\,R_{\oplus} < R < 13\,R_{\oplus}$ around young stars with ages <100 Myr. We can also see this feature in Figure 11.

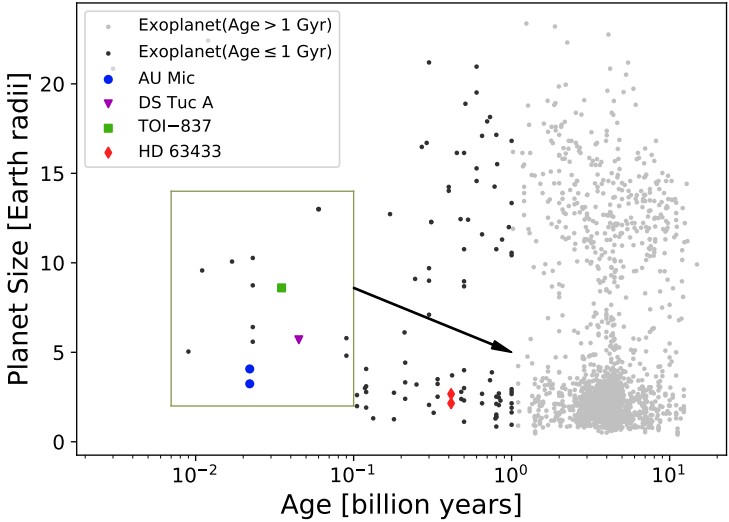

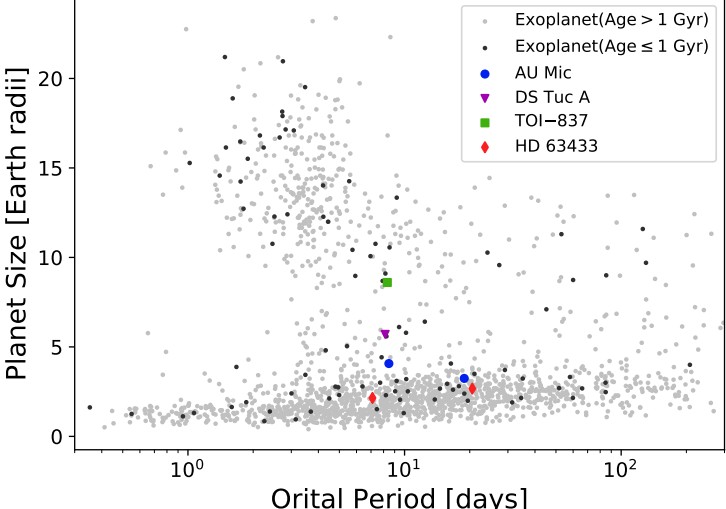

**Figure 11.** The distribution of young planets from this study. **Top panel**: Planet radii versus ages. **Bottom panel**: Planet radii versus orbital periods. We also show known planets from the NASA exoplanet archive for comparison. The ages, radii and periods are taken from the NASA Exoplanet Archive on 3 October 2022. The known planets with ages greater or smaller than 1 Gyr are marked using grey and black dots respectively. To minimize the influence of outliers, our selection of known planet sample has been restricted to those with a fractional radii measurement error of less than 30%.

One possible explanation raised by Bouma et al. [2] for this feature other than selection bias is that sub-100 Myr planets that are currently enveloped with primordial H/He atmospheres will become sub-Neptune-sized planets after undergoing atmospheric escape caused by photoevaporation or core-powered mass-loss [86–89]. We use a square and an arrow in Figure 11 to point out this possible evolution trend, where bigger young planets may become smaller as they become older. In the bottom panel of Figure 11, we separate the planets by the age of 1 Gyr and compare their distribution in the period-radii space. It seems they largely overlap with each other, which implies that most of the evolution effects may have occurred before the planetary age of 1 Gyr. However, the current sample of young planets is still too small to give a conclusive picture. In the future, a larger young planet

sample from an unbiased survey down to sub-Neptunes, like the one from Yee et al. [90], is critical to rule out a possible selection effect.

## 5. Conclusions

We have carried out a systematic search for short-period transiting exoplanets among 1075 stellar members of a young open cluster/association using the 2-min cadence TESS survey data. We find six planetary candidates out of 1075 targets, among which one is a possible hot Jupiter. We also find ten false positive signals, which are most likely eclipsing binaries or background variable stars. From this study, we can put a 95% confidence upper limit on the hot Jupiter occurrence rate orbiting stars in young stellar associations at <5.1%. To put a more tight constraint on the occurrence rate of hot Jupiter in young stellar associations, a much larger sample from a high-cadence, high-precision young transiting exoplanet survey from TESS, PLATO, or ET2.0 is needed in the future.

**Author Contributions:** Conceptualisation, B.M.; methodology, B.M. and Y.W.; validation, C.C. and Y.W.; investigation, Y.F., Y.W. and C.C.; writing—original draft preparation, Y.F. and B.M.; writing—review and editing, B.M. and C.C. All authors have read and agreed to the published version of the manuscript.

**Funding:** We acknowledge financial support from the National Key R&D Program of China (2020YFC 2201400), NSFC grant 12073092, 12103097, 12103098, the science research grants from the China Manned Space Project (No. CMS-CSST-2021-B09, B12), Guangzhou Basic and Applied Basic Research Program (202102080371), the Fundamental Research Funds for the Central Universities, Sun Yat-sen University.

**Data Availability Statement:** The data underlying this article were collected by the TESS mission, which are publicly available from the Mikulski Archive for Space Telescopes (MAST). The derived data generated in this research will be shared on reasonable request to the corresponding author.

**Acknowledgments:** We want to thank all the referee for his/her precious time in reviewing our paper and providing valuable comments. This paper includes data collected by the TESS mission, which are publicly available from the Mikulski Archive for Space Telescopes (MAST). Funding for the TESS mission is provided by NASA's Science Mission directorate. This research has made use of the SIMBAD database, operated at CDS, Strasbourg, France.

**Conflicts of Interest:** The funders had no role in the design of the study; in the collection, analyses, or interpretation of data; in the writing of the manuscript; or in the decision to publish the results.

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
