# Peer review of "Constraining Young Hot Jupiter Occurrence Rate in Stellar Associations Using 2-min Cadence TESS Data"

_universe, doi:10.3390/universe9040192_

Round 1

Reviewer 1 Report

In this manuscript, the authors performed a search for young planets in young open clusters and associations using the 2-mins cadence TESS survey data, and compared to previous studies. The authors then use Monte Carlo simulations to derive an upper limit on the hot Jupiter occurrence rate orbiting young stars. Once my comments are considered, I would be pleased to recommend acceptance of the paper.

My major comments on the draft are included below:

1. Figure 1: the color-magnitude diagram. I assume the stars in this survey have quite different distances, interstellar extinctions, and reddenings. Then plotting the isochrone on the same figure for all these stars seem problematic to me: 1) Which distance modulus did the authors use to shift the isochrone to an apparent G magnitude; 2) Did the authors consider interstellar extinction and reddening into consideration. Without taking these into consideration, it seems more like deliberately moving the isochrones to let the stars fall in between 10-100 Myr. I would recommend the authors to use the absolute magnitude (M_G) diagram (compute M_G for each of the stars using their distances from Gaia), and explain in more details on how they deal with reddening and interstellar extinctions.

The detection limits is one of the most important results in this paper, however, there are little text in this paper that explains how they carried out the Monte Carlo simulations. This could be explained in more details.

And a few minor comments:

line 3: candence -> cadence

Figure 11 caption: left panel -> top panel; right panel -> bottom panel

Reviewer 2 Report

Dear Authors,

Thank you for submitting your paper on the occurrence rate of hot Jupiters around young stars. The paper raises an exciting question that has the potential to contribute significantly to our understanding of hot Jupiter formation. However, after reading the paper, I have some concerns about its robustness and would like you to address them before publishing the paper.

Firstly, while the paper focuses on the hot Jupiter occurrence rate around young stars, the introduction does not adequately highlight this motivation. Please revise the introduction to convey the importance of the research question and how it relates to the hot Jupiter formation.

Secondly, I have some concerns about how the members of young clusters were selected for the study. The paper does not provide sufficient detail on how you picked these members or how you avoided including incorrect members or missing correct ones. 

Thirdly, to determine the occurrence rate of hot Jupiters, it is necessary to use systemic and automatic detection methods rather than manual inspection. The paper claims that the authors can distinguish contamination from nearby stars by eye, but it is not clear how this was accomplished. Additionally, second eclipse, and the depth of odd and even transits can be well distinguished by an automatic pipeline. 

Fourthly, the paper does not provide enough detail on the injection recovery test. This lack of information makes it difficult to judge the correctness of the work. 

Lastly, the overall results are based on two very uncertain values: the detection of one hot Jupiter and 1075 stars in young clusters. Any false-negative detection for the hot Jupiter or both false-negative and positive detections for the stars in the young clusters would significantly impact the results. 

In summary, I appreciate the contribution of your paper to the field of exoplanet research. However, to ensure the robustness of the results, please address the concerns raised above before publishing the paper.

Reviewer 3 Report

==============

Referee Report

==============

I read the paper "Constraining Hot Jupiter Occurrence Rate Using 2-mins Cadence TESS Data" by Yuanquing Fang et al., who analysed the TESS light curves of members of almost 30 stellar associations, open clusters and moving groups, in order to search for transiting planet candidates around young stars. The authors used the result of this investigation to put some constrain on the frequency of hot Jupiters around young stars. 

This topic is one of the key questions in the exoplanetary field, since it can help in constraining migration and evolution scenarios of the planetary systems, a still debated issue. However, after the presentation and validation/rejection of the planet candidates the authors detected with the TESS data, the methodology used to evaluate the frequency of hot Jupiters in those associations is omitted.

For this reason, the paper cannot be accepted in this stage, and I invite the authors to provide more details on their analysis.

============

Major issues

============

My main concern regards the description of the frequency analysis, which is described in a few lines in the "Discussion" section. This is the focus of the paper and the methodology should be reported with details (e.g. the efficiency of the transit detection, how the injection/retrival simulations worked - I guess they use this kind of approach - and so on...).

Moreover, the results are compared with similar works by Brucalassi et al. 2017 on M67 (3-5 Gyr) and Hartman et al. 2009 on M37 (8 Gyr), which are significantly older than the clusters considered in this work. A more recent result using TESS data for a large sample of members in open clusters and young association is presented e.g. by Nardiello et al. 2020, MNRAS 495, 4924. In that work, the rate is ~ 0.2%.

Finally, the frequency is evaluated only for members in open clusters and associations, while the field stars are not taken into account. In the literature there are examples of young stars hosting transiting planets that do not belong to stellar associations (e.g. Zhou et al. 2021, Desidera et al. 2023, ....). The authors should then claim that their frequency rate is only related to clusters and not to the whole young population.

============

Minor issues

============

The title should be modified in something like "Constraining Young Hot Jupiter Occurrence Rate Using 2-mins Cadence TESS Data in stellar associations", since the paper is focused to evaluate the frequency of young hot Jupiters in clusters/associations, otherwise it seems to be related to the global population of hot Jupiters.

Sect. 2

- Acronyms like SNR, QLP should be explained.

- Table 1: The footnote should be the label of the Table, moreover, there is no matching between the references in the note and the Table.

Sect. 3

- Please, define BLS, TLS and SDE_TLS (if possible, with references)

- Table 2: is the same as Table 1

- Table 4: The mass reported for DS Tuc A b was not measured by Newton et al., but by Benatti et al. 2021, A&A 650A, 66. Moreover, it is an upper limit, so it should be written " < 0.045". Similarly, the mass of TOI-837 is actually an upper limit, so it should be reported as "< 1.2".

- Figure 3 and 4 should be switched.

- The authors report that the "raw" TESS light curves of Au Mic, DS Tuc A and HD63433 show photometric modulations likely due to the stellar activity. It could be useful to include the typical values of amplitude and period of such modulations, so the reader can compare them with the orbital periods and transit dephts of the detected planets. 

- Fig. 6: uniform the label with the plots.

Sect. 4

- Figure 11: Left -> Upper; Right -> Lower. Are the radii of the known population (i.e. grey and black dots) robustly measured? I mean, did the authors set a limit on the radii uncertainty before plotting or did they considered all the available radii without considering it? This information is useful since the Figure should compare the young planets population with a robust distribution of older planets without outliers due to wrong/non robust estimation of their radii.

- Lines 191-192: Better to motivate this sentence. Something like: "...will tend to be lower in young clusters, since the time scale for this process requires hundreds of Myr up to 1 Gyr for the completion (ref)". A possible reference can be Chatterjee et al. 2008, ApJ, 686, 580.

- As previously stated, the methodology for the frequency evaluation deserves more than three lines. Please, provide more details on the analysis performed to evaluate the frequency rate of young hot Jupiters.

A few typos:

Typo in Figure 2: "Peroidogram"

l. 66: observers -> observes

l. 136: Is -> It

l. 155: an -> a

l. 189: formatino -> formation

l. 235: right -> lower

Round 2

Reviewer 2 Report

I appreciate the effort you have put into improving the paper, and I believe it is now ready for publication. 

Reviewer 3 Report

I'd like to thank the Authors for the careful revision of the manuscript and the consideration of my suggestions. 

I think that the manuscript is now ready for publication since all the major/minor points I raised are now fixed.